# Selection of forage grasses for cultivation under water-limited conditions using Manhattan distance and TOPSIS

**Bruno Rodrigues de Oliveira**[1]*, **Marco Aparecido Queiroz Duarte**[2], **Alan Mario Zuffo**[3], **Fábio Steiner**[4], **Jorge González Aguilera**[1,4], **Alexson Filgueiras Dutra**[5], **Francisco de Alcântara Neto**[6], **Marcos Renan Lima Leite**[6], **Nágila Sabrina Guedes da Silva**[6], **Eliseo Pumacallahui Salcedo**[7], **Luis Morales-Aranibar**[7], **Richar Marlon Mollinedo Chura**[8], **Roger Ccama Alejo**[8], **Wilberth Caviedes Contreras**[9]

1 Pantanal Editora, Nova Xavantina-MT, Brasil, 2 Departamento de Matemática, Universidade Estadual de Mato Grosso do Sul (UEMS), Unidade de Cassilândia, Cassilândia-MS, Brasil, 3 Departamento de Agronomia, Universidade Estadual do Maranhão (UEMA), Campus Balsas, Balsas-MA, Brasil, 4 Departamento de Agronomia, Universidade Estadual de Mato Grosso do Sul (UEMS), Unidade de Cassilândia, Cassilândia-MS, Brasil, 5 Instituto Federal do Tocantins, Pedro Afonso-TO, Brasil, 6 Programa de Pós-Graduação em Agronomia, Universidade Federal do Piauí Teresina, Piauí, Brasil, 7 Departamento de Ingeniería Civil y Ciencias Básicas, Universidad Nacional Intercultural de Quillabamba (UNIQ), Cusco, Perú, 8 Departamento Académico de Ciencias Físico Matemáticas, Universidad Nacional del Altiplano—Puno, Puno, Perú, 9 Departamento Académico de Ciencias Básicas, Universidad Nacional Amazónica de Madre de Dios (UNAMAD), Madre de Dios, Perú

* bruno@editorapantanal.com.br

**Data Availability Statement:** All relevant data are within the paper and its Supporting Information files, and are publicly available from the Github

## Abstract

Extreme weather events, such as severe droughts, pose a threat to the sustainability of beef cattle by limiting the growth and development of forage plants and reducing the available pasture for animals. Thus, the search for forage species that are more tolerant and adapted to soil water deficit conditions is an important strategy to improve food supply. In this study, we propose utilizing the mathematical concept of the Manhattan distance to assess the variations in the morphological variables of tropical forage grasses under water-limited conditions. This study aimed to select genotypes of tropical forage grasses under different water stress levels (moderate or severe) at this distance and the Technique for Order Preference by Similarity to Ideal Solution (TOPSIS). Nine varieties from five species were examined. Forage grasses were grown in 12-L pots under three soil irrigation regimes [100% pot capacity–PC (well-irrigated control), 60% PC (moderate drought stress), and 25% PC (severe drought stress)] with four replicates. Drought stress treatments were applied for 25 days during the forage grass tillering and stalk elongation phases. After exposure to drought stress, the growth and morphological traits of forage plants were evaluated. The results show that the use of the Manhattan distance combined with TOPSIS helps in the genotypic selection of more stable tropical forage grass varieties when comparing plants exposed to moderate and severe drought conditions in relation to the nonstressful environment (control). The 'ADR 300', 'Pojuca', 'Marandu', and 'Xaraés' varieties show greater stability when grown in a greenhouse and subjected to water stress environments. The selected forage varieties can be used as parents in plant breeding programs, allowing us to obtain new drought-resistant genotypes.

repository (https://github.com/brunobro/selection-of-forage-grasses-for-cultivation-under-water-limited-conditions).

**Funding:** The authors received no specific funding for this work. The funders had no role in study design, data collection and analysis, decision to publish, or preparation of the manuscript.

**Competing interests:** The authors have declared that no competing interests exist.

## Introduction

Pastures play a crucial role in beef cattle farming in Brazil owing to their low production costs and high practicality in providing a consistent supply of food for cattle. Brazil is the largest exporter and second-largest producer of beef in the world. According to the Municipal Livestock Survey of the Brazilian Institute of Geography and Statistics, Brazil reached 224.6 million heads of cattle in 2021 [1]. This year, the pasture area was estimated at 158 million hectares [2,3]. Therefore, as the production of beef cattle is based on pasture systems, the quality and biomass production potential of forage plants are essential for Brazilian livestock activity [4].

Among the main varieties of tropical forage used and marketed in Brazil, the genera *Urochloa* sp. (syn. *Brachiaria* sp.), *Panicum* sp., *Cynodon* sp., and *Paspalum* sp. stand out due to their high forage production potential in the tropical Brazilian Cerrado region [5–7]. These tropical forage grasses have been widely cultivated in Brazil due to their excellent production stability, nutritional quality, and wide adaptability to the country's different edaphoclimatic conditions [5,7,8]. However, each forage species has a different potential for forage production, which is generally dependent on the genetic and morphological characteristics of the plants and climatic conditions [7,9].

Although palisade grass [*Urochloa brizantha* (Hochst. Ex A. Rich.) R. D. Webster] and ruzigrass [*U. ruziziensis* (R. Germ. & C. M. Evrard) Crins] have been described as drought-tolerant tropical forage grasses, and low soil water availability often limits forage plant development and forage supply [5,7]. Therefore, the sustainability of beef cattle production in many regions of Brazil is at risk due to the scarcity of soil water, especially during the dry season (i.e., the period between April and September with poor forage supply for beef cattle).

Water stress affects the productivity and quality of tropical forage grasses. Tropical forage grasses are crucial in livestock production, mainly due to their role in soil and ecosystem conservation in the subtropics and tropics [6,8]. Water stress can reduce the growth and consequently the biomass and nutritional value of tropical forage grasses [5,7,10], as well as increase their susceptibility to pests and diseases [11,12]. Drought stress triggers changes in the physiological and biochemical responses of plants, including photosynthesis, water relations, antioxidant defense, and osmotic adjustment [10,13,14]. Morphological and biochemical changes were described by Zhang et al. [15] as strategies to address water stress in tropical forage grasses. Understanding the mechanisms that allow adaptation and tolerance to water stress in tropical forage grasses can help to improve their management and the selection of superior genotypes in forage breeding to promote better performance in conditions of water scarcity.

Multivariate analysis methods are employed in many areas, including agriculture, where they can help to choose the most robust species of tropical forage grasses under stress conditions, for example. Among the available methods, there are several proposals that aim to evaluate the adaptation and tolerance of plant species exposed to stressful environments [7,16]. The use of a new approach employing the distance (or similarity) of a vector space through the Manhattan distance and the Technique for Order Preference by Similarity to Ideal Solution (TOPSIS) method allowed the selection of soybean cultivars subjected to saline stress [16]. This method allowed the selection of forage grass genotypes in water stress environments, with superior performance and low temporal variability. This is a desirable trait, as it is beneficial for the selection of forages resistant to stress conditions, which contributes to obtaining better pastures for raising beef cattle.

The hypothesis of this research is that the selection method from [16] can be used to simultaneously select tropical forage grass species under moderate and severe drought stress conditions. The authors of the method have suggested its applicability to other plant species and contexts due to its general purpose, which remains an avenue for future exploration. Thus, applying this method, we intend to select the best forage grass varieties according to the

morphological characteristics of the plants exposed to nonstressful (control) and stressful conditions (moderate and severe drought stress) obtained in the experiments carried out by Zuffo et al. [7]. The study collected data on several morphological variables of the plants, including plant height (PH), number of tillers (NT), number of green leaves (NGL), leaf area (LA), shoot dry matter (SDM), root dry matter (RDM), total dry matter (TMD), and root volume (RV). The Manhattan distances between the control and stressed samples were calculated using these variables. The TOPSIS method was then employed to derive a score to identify the cultivar that demonstrated the least deviation from the control sample and was considered most stable under drought stress conditions.

## Materials and methods

### Plant material and stress treatment

The experimental design was composed of a fully randomized block structure consisting of a 9 x 3 factorial arrangement and was replicated four times. Plants from nine varieties of tropical forage grasses, three varieties of *Urochloa brizantha* (Hochst. Ex A. Rich.) R.D. Webster ('BRS Piatã', 'Marandu' and 'Xaraés'), three varieties of *Panicum maximum* Jacq. ('Aruana', 'Mombaça' and 'Tanzânia'), a cultivar of *Pennisetum glaucum* (L.) R. Br. ('ADR 300'), a cultivar of *Urochloa ruziziensis* (R. Germ. & C.M. Evrard) Crins ('Comum') and a cultivar of *Paspalum atratum* Swallen ('Pojuca') were grown in 12 L plastic pots filled with high-fertility sandy loam soil under greenhouse conditions. The environmental conditions inside the greenhouse during the experiment were controlled and maintained at an average air temperature of 26.0˚C (±2˚C), average air relative humidity of 70% (±4%), and midday photosynthetic photon flux density of 630 μmol m$^{-2}$ s$^{-1}$ (±180 μmol m$^{-2}$ s$^{-1}$).

Field capacity, or its equivalent for pot experiments, 'pot capacity', was determined under free-draining conditions as previously recommended by [15]. For this, a water content decrease rate of 0.1 g kg$^{-1}$ day$^{-1}$ was used, as proposed by Casaroli and Lier [17], and the soil moisture content (SMC) in the pot capacity (PC) was 218 g kg$^{-1}$. Ten viable seeds were sown at a depth of 2.0 cm, and following five days of emergence, the seedlings underwent thinning procedures resulting in the retention of only two plants per pot. Each plot consisted of a vase with two plants. In the fertilization management, urea at 50 mg kg$^{-1}$ was used as a source of nitrogen (N); for phosphorus (P), simple superphosphate at 300 mg kg$^{-1}$; for potassium (K), potassium chloride at 150 mg kg$^{-1;}$ and sulfur (S) or agricultural gypsum at 30 mg dm$^{-1}$. Regarding micronutrients, copper sulfate was used as a source of Cu, and zinc sulfate was used to provide Zn, both at a dose of 2 mg kg-1$^{-1}$. All of the species were given fertilization treatment using a urea-based solution containing 80 mg kg$^{-1}$, exactly 30 days after the plants first emerged. Information on the soil chemical properties and agronomic characteristics of the investigated tropical forage grass varieties was described by Zuffo et al. [7].

Until 40 days after sowing, the SMC was maintained at PC (218 g kg$^{-1}$) with daily irrigation. Afterward, the experiment was divided into three groups of soil irrigation regimes [100% of PC (well-irrigated control), 60% of PC (moderate drought stress), and 25% of PC (severe drought stress)] with four replicates. Drought stress categories were proposed in [18], whereby moderate and severe water stresses were achieved by maintaining soil moisture at 20%-30% and 50%-60% PC, respectively, in pots. There are also other authors who used these PC values for moderate and severe drought [7].

Drought stress treatments were applied for 25 days during the forage grass tillering and stalk elongation phases. The SMC was monitored daily at 9:00 a.m. and 3:00 p.m. using the gravimetric method described in [18], and the SMC was adjusted by adding water after weighing the pot.

## Quantification of morphological traits

After the 25th day of exposure to water stress, the forage plants were harvested, and plant height (PH), number of tillers (NT), number of green leaves (NGL), leaf area (LA), and volume of root (RV) were measured. The plants were separated into leaves, stems and roots, dried in a forced-air oven at 65°C for three days and then weighed. The aerial part dry matter (MSD) was obtained from the sum of the dry matter of the leaves and stems. Total dry matter (TDM) was obtained from the sum of all parts of the plant (leaves, stems and roots).

The PH was determined using a metal tape measure, considering the distance from the soil surface to the +1 leaf (Kuijper's leaf numbering system [19]). The NT determination corresponded to the total number of plants in each pot. For NGL, only green leaves were considered (fully expanded with a minimum green area of 20%, counted from the +1 leaf). The LA was estimated using the equation proposed by Benincasa [20]: LA = [(LAs × LTDM)/DMs], where LA is the leaf area, LAs is the leaf area of the collected sample, LTDM is the leaf total dry matter, and DMs is the dry matter of the collected sample. The RV was determined using a 1,000 mL graduated cylinder by the water displacement method.

Since the objective of the present study is to employ the selection method proposed in [16], some statistical analyses of the experiments are not displayed here. They can be found in [7]. However, it is important to mention that the data were submitted to statistical hypothesis tests to verify the normality of residues using the Shapiro–Wilk test with $p > 0.05$ and homogeneity of variances using the Levene test with $p > 0.05$. Afterwards, analysis of variance (ANOVA) was applied to the data. For significant values, means were also compared using the Scott–Knott test with a confidence level of 0.05 [7].

## Similarity through Manhattan distance

In this work, a vector is considered to be a set $x = (x_1, x_2, \ldots, x_n)$ containing information about an observed phenomenon, where $x_i$, $i = 1, \ldots, n$, relates to the information on every feature (variable) $i$. Therefore, $x$ is a data vector.

The distance function of two vectors $x = (x_1, x_2, \ldots, x_n)$ and $y = (y_1, y_2, \ldots, y_n)$ represented by $d(x,y)$ must satisfy the following requirements: a) $d(x,y) > 0$; b) $d(x,x) = 0$; c) $d(x,y) = d(y,x)$; and, given a third vector $z$, d) $d(x,y) + d(y,z) \geq d(x,z)$ [21]. The Minkowski distances are defined as $d(x, y, p) = \left( \sum_{i=1}^{n} |x_i - y_i|^p \right)^{1/p}$. For $p = 1$ and $p = 2$, we have Manhattan and Euclidean distances, respectively. Euclidean distance is the concept of distance that we use in everyday life. It is the distance in a straight line. On the other hand, the Manhattan distance is intuitively understood as the distance traveled in a city divided by parallel and perpendicular streets. Two objects that have a small distance in an n-dimensional space are more similar to each other than objects that have greater distances when we consider that each dimension represents a variable that describes some characteristic of that object [22].

In [23], the authors understood that the Manhattan distance is the most adequate to verify the difference between spaces of higher dimension. Therefore, this distance was chosen [16] to verify which of the samples of the varieties under stress are more (or less) distant from those samples of the control varieties. The TOPSIS method (described in the following section) uses Euclidean distance.

Since the measured variables of forage grass varieties are on different scales, [14] proposed using a preprocessing step (normalization). It consists of dividing the variables by their maximum value, that is, $\tilde{x}_i = x_i / max(x_i)$, where $x_i$ is the value of the variable for the $i$-th sample and $\tilde{x}_i$ is the respective normalized value. In this way, there is a guarantee that $\tilde{x}_i$ is in the interval between 0 and 1. In this way, the variables become dimensionless. It obviously does not affect the results because the objective is to calculate only the scores by the TOPSIS method. Therefore, the dimension of the variable is irrelevant.

## Technique for Order Preference by Similarity to Ideal Solution (TOPSIS)

TOPSIS is commonly used in decision-making problems in the most diverse areas. It is a multicriteria analysis method designed to rank alternatives by evaluating the criteria to which they are subject. This method considers both the distances between the alternatives and the ideal solutions and the distances between the alternatives themselves. The algorithm starts by defining the criteria to be evaluated and then applies normalization to the data so that the criteria have the same weight in the evaluation. Next, decision and weighting matrices are constructed to compare the alternatives concerning the criteria. From these matrices, the distances between each alternative and the ideal positive (highest value of each criterion) and negative (lowest value of each criterion) solutions are calculated, and the distances between the alternatives are also calculated. From these distances, the similarity values of each alternative with the positive and negative ideal solutions are calculated, and the relative performance of each alternative is calculated as the ratio of these similarities through a score. Finally, the alternatives are ranked according to their relative performance, in descending order, to provide an order of preference of alternatives [24,25].

The TOPSIS method can be employed in six steps [26]. Let $X = (x_{ij})_{m \times n}$ be a decision matrix with $m$ alternatives and $n$ criteria, where $x_{ij}$ is the value of alternative $i$ concerning criterion $j$ [16]:

Step 1. Normalize the decision matrix: $r_{ij} = x_{ij} / \sum_{i=m}^{m} \sum_{j=1}^{n} x_{ij}^2$. This ensures that each criterion has the same relative weight;

Step 2. Given a criteria vector (weight) $w = [w_1, w_2, \ldots, w_n]$, calculate the product $v_{ij} = r_{ij} w_j^T$ to obtain a weighted normalized decision matrix such that the sum of the weights is equal to one, i.e., $\Sigma w_j = 1$, for $j = 1, 2, \ldots, n$;

Step 3. Determine the best alternative $A_b$ (positive-ideal solution) and worst alternative $A_w$ (negative-ideal solution) as $A_{wj} = max_{i=1}^{m} v_{ij}$ and $A_{bj} = min_{i=1}^{m} v_{ij}$, respectively, for each criterion. Note that we are dealing with minimization criteria because the shorter the distance, the better the forage grass cultivar suffered the effects of stress;

Step 4. Calculate the Euclidean distance between alternative $i$ and the worst ($A_w$) and best ($A_b$) alternatives, defined in the previous step: $S_i^w = \sqrt{\sum_{i=1}^{m} (v_{ij} - A_{wj})^2}$ and $S_i^b = \sqrt{\sum_{i=1}^{m} (v_{ij} - A_{bj})^2}$, respectively;

Step 5. Calculate the relative closeness from each alternative $i$ to the worst alternative: $C_i = S_i^w / (S_i^w + S_i^b)$.

Finally, $C_i$ provides a score that is used to rank the alternatives in order of importance relative to the analyzed criteria. The higher its value is, the closer this alternative is to the ideal solution. In relation to the present research, the alternatives are forage grass varieties, while the criteria are Manhattan distances (control/moderate and control/severe) in the environment comparison.

Other studies also used TOPSIS for genotype selection but without using distance measures as performed in this work. Among these, we mention the selection of genotypes for plant bioassays, which uses ANOVA together with TOPSIS [27]; the cultivar selection test platform for which the authors also mention the use of TOPSIS together with other techniques [28]; and the selection of chickpea cultivars based on their functional properties [29].

## Results

A preprocessing step was performed before calculating the distance measure. This step consists of normalizing the data for each variable so that they are within the range[0,1]. Normalized

**Table 1. Raw values in the control and stressed environments for all varieties.**

| Cultivar | PH (cm) | NT (unit) | NGL (unit) | RV (mm³) | LA (cm²) | SDM (g) | RDM (g) | TDM (g) |
|---|---|---|---|---|---|---|---|---|
| | | | | Control | | | | |
| ADR 300 | 154.1 | 5.1111 | 26.2222 | 53.3333 | 1.6492 | 27.96 | 5.3778 | 33.3378 |
| Aruana | 72.6667 | 18.3333 | 75.1667 | 113.0 | 28.8678 | 40.5383 | 17.5322 | 58.0706 |
| BRS Piatã | 74.0 | 13.5556 | 39.8889 | 88.8889 | 16.883 | 32.2867 | 23.3322 | 55.6189 |
| Comum | 47.6667 | 27.5556 | 119.4444 | 154.4444 | 31.6483 | 34.0044 | 22.8522 | 56.8567 |
| Marandu | 62.6667 | 14.6667 | 41.3333 | 106.6667 | 20.8522 | 31.2622 | 19.48 | 50.7422 |
| Mombaça | 79.6667 | 18.7222 | 50.6667 | 162.7778 | 28.6113 | 40.9739 | 28.3589 | 69.3328 |
| Pojuca | 62.0 | 22.5556 | 195.4444 | 64.4444 | 10.7532 | 27.3678 | 7.0533 | 34.4211 |
| Tanzânia | 68.6667 | 14.7778 | 51.8889 | 133.3333 | 23.3387 | 35.3278 | 20.9789 | 56.3067 |
| Xaraés | 62.0 | 12.7778 | 36.0 | 94.0 | 21.0079 | 28.3933 | 14.0233 | 42.4167 |
| | | | | Moderate Drought Stress | | | | |
| ADR 300 | 124.5833 | 3.8889 | 21.1111 | 31.6667 | 1.3981 | 20.1389 | 5.1933 | 25.3322 |
| Aruana | 71.0 | 21.0556 | 47.2778 | 85.0 | 16.9843 | 25.6917 | 8.2911 | 33.9828 |
| BRS Piatã | 63.6667 | 10.2222 | 33.1111 | 64.4444 | 10.1803 | 20.9867 | 15.0178 | 36.0044 |
| Comum | 47.3333 | 28.8333 | 87.7222 | 85.0 | 18.9083 | 31.4794 | 14.4222 | 45.9017 |
| Marandu | 51.3333 | 13.0 | 31.5556 | 61.7778 | 17.5471 | 22.8 | 12.5256 | 35.3256 |
| Mombaça | 62.6667 | 15.2222 | 30.3333 | 71.1111 | 18.7396 | 28.9733 | 13.09 | 42.0633 |
| Pojuca | 52.3333 | 18.219 | 94.5048 | 22.0952 | 5.4618 | 14.3248 | 3.6481 | 17.9729 |
| Tanzânia | 63.6667 | 18.4444 | 32.1111 | 83.3333 | 21.1421 | 28.1878 | 15.0289 | 43.2167 |
| Xaraés | 56.0 | 11.8889 | 25.7778 | 58.8889 | 14.8102 | 20.2311 | 12.22 | 32.4511 |
| | | | | Severe Drought Stress | | | | |
| ADR 300 | 86.25 | 3.5556 | 15.3333 | 23.3333 | 1.3365 | 14.15 | 3.5556 | 17.7056 |
| Aruana | 71.3333 | 15.6667 | 37.8889 | 38.8889 | 11.9335 | 25.5922 | 1.255 | 26.8472 |
| BRS Piatã | 53.3333 | 8.6667 | 19.6667 | 38.8889 | 7.5927 | 14.8178 | 8.2122 | 23.03 |
| Comum | 43.3333 | 21.8889 | 50.7778 | 33.2222 | 11.9331 | 19.0378 | 7.0556 | 26.0933 |
| Marandu | 50.0 | 9.3333 | 24.3333 | 37.7778 | 9.8709 | 15.6256 | 7.5444 | 23.17 |
| Mombaça | 62.0 | 13.4444 | 22.6667 | 52.2222 | 11.9982 | 21.6222 | 8.2789 | 29.9011 |
| Pojuca | 52.6667 | 18.5556 | 70.3333 | 21.1111 | 4.3988 | 13.0289 | 4.2478 | 17.2767 |
| Tanzânia | 63.6667 | 13.4444 | 32.1111 | 58.8889 | 12.8901 | 21.4278 | 12.2811 | 33.7089 |
| Xaraés | 50.6667 | 8.6667 | 18.3333 | 32.8889 | 8.8465 | 13.8222 | 8.7889 | 22.6111 |

PH: Plant height; NT: Number of tillers; NGL: Number of green leaves; LA: Leaf area; SDM: Shoot dry matter; RDM: Root dry matter; TDM: Total dry matter; and RV: Root volume.

data are presented for each of the environmental conditions: well-irrigated control and moderate and severe water stress. Tables 1 and 2 show the raw data and normalized data, respectively.

The results of the variance analysis indicated that the soil water regime had a significant impact ($p < 0.01$) on all growth traits of forage grass (Table 3). Additionally, the interaction between soil water regime and cultivars had a significant influence ($p < 0.05$) on all growth traits of the plant, with the exception of the number of tillers and shoot dry matter. The notable interaction between the primary effects of cultivars and soil water regimes on the majority of morphological traits suggests that forage grasses exhibit different reactions when subjected to varying levels of soil water availability [7].

The Manhattan distances between the control samples and the samples from the drought stress environment are shown in Fig 1. These distances were calculated considering all variables. That is, each sample containing the measurements of the 8 variables represents a point

**Table 2. Normalized values in the control and stressed environments for all varieties.**

| Cultivar | PH (cm) | NT (unit) | NGL (unit) | RV (mm³) | LA (cm²) | SDM (g) | RDM (g) | TDM (g) |
|---|---|---|---|---|---|---|---|---|
| **Control** | | | | | | | | |
| ADR 300 | 1.0000 | 0.1855 | 0.1342 | 0.3276 | 0.0521 | 0.6824 | 0.1896 | 0.4808 |
| Aruana | 0.4716 | 0.6653 | 0.3846 | 0.6942 | 0.9121 | 0.9894 | 0.6182 | 0.8376 |
| BRS Piatã | 0.4802 | 0.4919 | 0.2041 | 0.5461 | 0.5335 | 0.7880 | 0.8227 | 0.8022 |
| Comum | 0.3093 | 1.0000 | 0.6111 | 0.9488 | 1.0000 | 0.8299 | 0.8058 | 0.8201 |
| Marandu | 0.4067 | 0.5323 | 0.2115 | 0.6553 | 0.6589 | 0.7630 | 0.6869 | 0.7319 |
| Mombaça | 0.5170 | 0.6794 | 0.2592 | 1.0000 | 0.9040 | 1.0000 | 1.0000 | 1.0000 |
| Pojuca | 0.4023 | 0.8185 | 1.0000 | 0.3959 | 0.3398 | 0.6679 | 0.2487 | 0.4965 |
| Tanzânia | 0.4456 | 0.5363 | 0.2655 | 0.8191 | 0.7374 | 0.8622 | 0.7398 | 0.8121 |
| Xaraés | 0.4023 | 0.4637 | 0.1842 | 0.5775 | 0.6638 | 0.6930 | 0.4945 | 0.6118 |
| **Moderate Drought Stress** | | | | | | | | |
| ADR 300 | 1.0000 | 0.1349 | 0.2234 | 0.3725 | 0.0661 | 0.6397 | 0.3456 | 0.5519 |
| Aruana | 0.5699 | 0.7303 | 0.5003 | 1.0000 | 0.8033 | 0.8161 | 0.5517 | 0.7403 |
| BRS Piatã | 0.5110 | 0.3545 | 0.3504 | 0.7582 | 0.4815 | 0.6667 | 0.9993 | 0.7844 |
| Comum | 0.3799 | 1.0000 | 0.9282 | 1.0000 | 0.8943 | 1.0000 | 0.9596 | 1.0000 |
| Marandu | 0.4120 | 0.4509 | 0.3339 | 0.7268 | 0.8300 | 0.7243 | 0.8334 | 0.7696 |
| Mombaça | 0.50300 | 0.5279 | 0.3210 | 0.8366 | 0.8864 | 0.9204 | 0.8710 | 0.9164 |
| Pojuca | 0.4201 | 0.6319 | 1.0000 | 0.2599 | 0.2583 | 0.4551 | 0.2427 | 0.3916 |
| Tanzânia | 0.5110 | 0.6397 | 0.3398 | 0.9804 | 1.0000 | 0.8954 | 1.0000 | 0.9415 |
| Xaraés | 0.4495 | 0.4123 | 0.2728 | 0.6928 | 0.7005 | 0.6427 | 0.8131 | 0.707 |
| **Severe Drought Stress** | | | | | | | | |
| ADR 300 | 1.0000 | 0.1624 | 0.218 | 0.3962 | 0.1037 | 0.5529 | 0.2895 | 0.5252 |
| Aruana | 0.8271 | 0.7157 | 0.5387 | 0.6604 | 0.9258 | 1.0000 | 0.1022 | 0.7964 |
| BRS Piatã | 0.6184 | 0.3959 | 0.2796 | 0.6604 | 0.589 | 0.5790 | 0.6687 | 0.6832 |
| Comum | 0.5024 | 1.000 | 0.7220 | 0.5642 | 0.9258 | 0.7439 | 0.5745 | 0.7741 |
| Marandu | 0.5797 | 0.4264 | 0.3460 | 0.6415 | 0.7658 | 0.6106 | 0.6143 | 0.6874 |
| Mombaça | 0.7188 | 0.6142 | 0.3223 | 0.8868 | 0.9308 | 0.8449 | 0.6741 | 0.8870 |
| Pojuca | 0.6106 | 0.8477 | 1.0000 | 0.3585 | 0.3413 | 0.5091 | 0.3459 | 0.5125 |
| Tanzânia | 0.7382 | 0.6142 | 0.4566 | 1.0000 | 1.0000 | 0.8373 | 1.0000 | 1.0000 |
| Xaraés | 0.5874 | 0.3959 | 0.2607 | 0.5585 | 0.6863 | 0.5401 | 0.7156 | 0.6708 |

PH: Plant height; NT: Number of tillers; NGL: Number of green leaves; LA: Leaf area; SDM: Shoot dry matter; RDM: Root dry matter; TDM: Total dry matter; and RV: Root volume.

**Table 3. Analysis of variance for morphological traits of forage grass cultivars under stress regimes [7].**

| Causes of Variation | Probability > F | | | | | | | |
|---|---|---|---|---|---|---|---|---|
| | PH (cm) | NT (unit) | NGL (unit) | RV (mm³) | LA (cm²) | SDM (g) | RDM (g) | TDM (g) |
| Forage cultivar (C) | <0.01 | <0.01 | <0.01 | <0.01 | <0.01 | <0.01 | <0.01 | <0.01 |
| Soil water regime (W) | <0.01 | <0.01 | <0.01 | <0.01 | <0.01 | <0.01 | <0.01 | <0.01 |
| C × W | <0.01 | 0.914 | <0.01 | <0.01 | 0.045 | 0.518 | 0.020 | 0.029 |
| CV (%) | 13.65 | 16.41 | 18.61 | 21.92 | 17.82 | 15.99 | 20.66 | 15.70 |

PH: Plant height; NT: Number of tillers; NGL: Number of green leaves; LA: Leaf area; SDM: Shoot dry matter; RDM: Root dry matter; TDM: Total dry matter; and RV: Root volume.

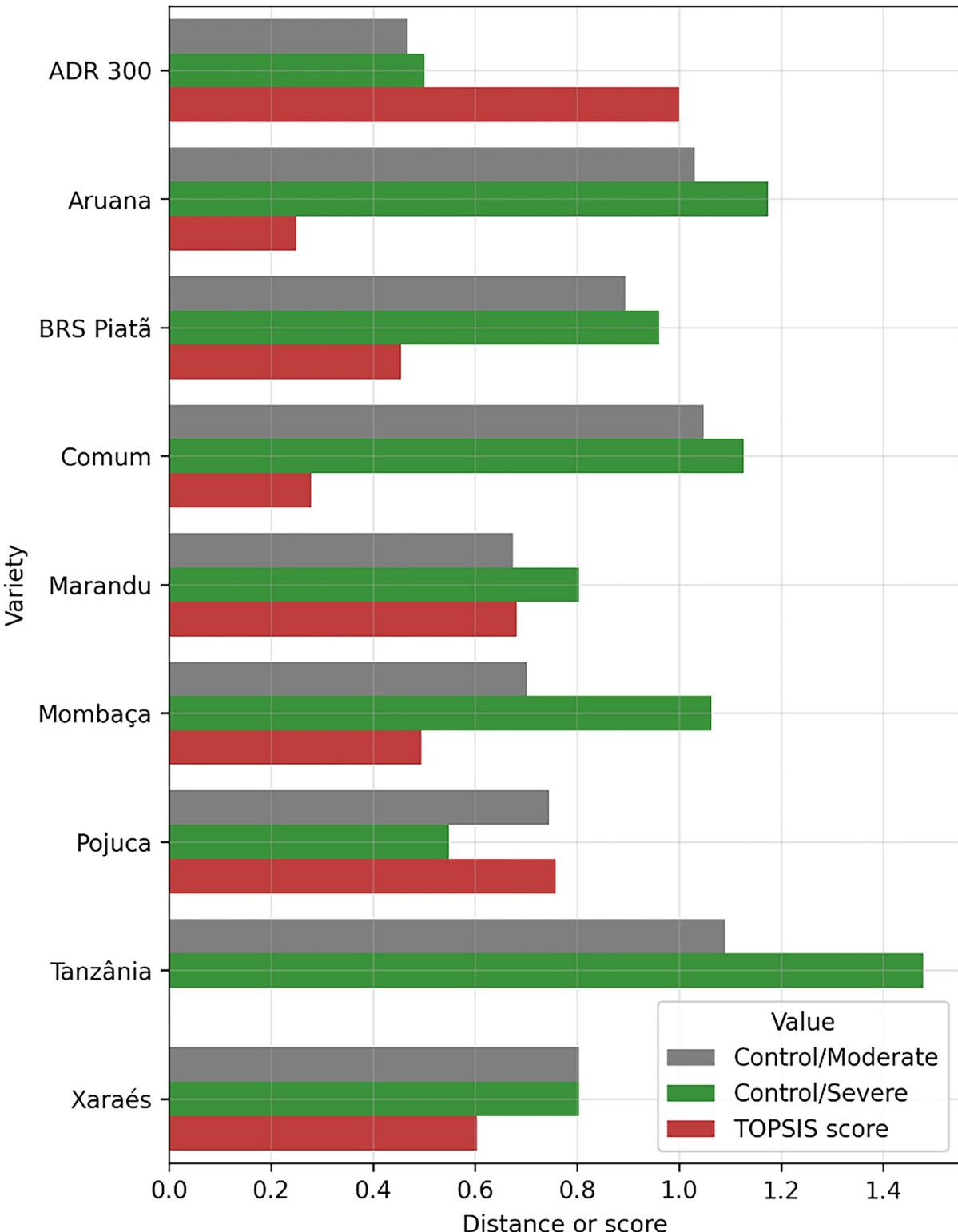

**Fig 1. Normalized Manhattan distances calculated between samples from the control and water stress environments and the TOPSIS score obtained using these distances as criteria for each forage grass cultivar.**

in an 8-dimensional space. Therefore, distances were calculated for these points. Additionally, the TOPSIS scores calculated using these distances as criteria and forage grass varieties as alternatives are shown. For the calculation of the scores, it was considered that both criteria have the same weight (maximum value of 0.5). Note in Fig 1 that for the 'Tanzânia' cultivar, the TOPSIS score is null, which is why its bar does not appear on the graph.

Criteria weights can influence the TOPSIS score (see step 2 Section "Technique for Order Preference by Similarity to Ideal Solution (TOPSIS)"). A certain cultivar of tropical forage grass can have very different distance measurements when comparing drought stress environments with well-irrigated controls. That is, in moderate stress conditions, the distance may be small, while in severe drought stress conditions, it may present a large distance, and vice versa. This is evident in the bar graph in Fig 1, where the control/severe values were in most cultivars superior to the control/moderate values, with the exception of the 'Pojuca' cultivar, which manifested an inverse behavior, and the 'Xáraes' cultivar, which showed the same behavior in both evaluated conditions.

Therefore, it is necessary to verify whether some alterations in the weights of the criteria impact the selection of forage grass varieties and to verify how much they impact this selection. Therefore, we vary the weights of the criteria (considering that their sum must be 1), and we recalculate the TOPSIS score to verify which will be the new selection order for the forage grass varieties. The results are listed in Table 4, except when considering the equal weights, as this result had already been expressed in Fig 1.

Considering the TOPSIS score (Fig 1), the four best (most stable) forage grass varieties are 'ADR 300', 'Pojuca', 'Marandu', and 'Xaraés'. We observed that this selection is slightly changed when different weights of the TOPSIS criteria are considered, according to the results shown in Table 4. On the other hand, the four varieties of grass that suffered the greatest changes in the measured variables were 'Tanzânia', 'Aruana', 'Comum', and 'BRS Piatã'. Figs 2 and 3 show the comparisons between the values of the variables (without normalization) of these forage varieties to visually verify how much these values are altered in drought stress conditions when compared with the values of the well-irrigated control.

In addition to the results presented in Figs 2 and 3, we present the percentage change values in Table 4. They are calculated by the following formula:

$$P = 100 \times (Control\ Value - Stress\ Value)/Control\ Value.$$

This calculation is made for each morphological trait and for moderate and severe drought stress conditions. We present the calculations for the top and worst four varieties selected by TOPSIS, considering equal weights for the criteria, i.e., these are those varieties selected according to Fig 1. Negative mean values indicate that there was an increase in the morphological characteristic in relation to the control environment. For the data shown in Table 5, only the varieties ('Tanzânia', 'Aruana' and 'Comum') considered among the four worst ones showed negative values for the NT variable when comparing Control/Moderate.

## Discussion

Multivariate data analysis is essential in experiments where many variables are measured, and the final decision depends on analyzing all these variables simultaneously [7,16]. We can see from the normalized data in Table 2 that the decision about which forage cultivar is less affected by water stress conditions is not a trivial task. This occurs because a certain variable may undergo drastic changes for a given cultivar, while other variables may change much less for the same cultivar. That is, there is no coherence between the changes in the morphological characteristics of forage plants [16].

**Table 4. Varieties selected by TOPSIS when criteria weights (Control/Moderate and Control/Severe distances) vary.**

| Weights | | Cultivar | TOPSIS score |
|---|---|---|---|
| Control/Moderate | Control/Severe | | |
| 0.1 | 0.9 | ADR 300 | 1.0000 |
| | | Pojuca | 0.9399 |
| | | Marandu | 0.6900 |
| | | Xaraés | 0.6880 |
| | | BRS Piatã | 0.5270 |
| | | Mombaça | 0.4250 |
| | | Comum | 0.3585 |
| | | Aruana | 0.3086 |
| | | Tanzânia | 0.0000 |
| 0.3 | 0.7 | ADR 300 | 1.0000 |
| | | Pojuca | 0.8665 |
| | | Marandu | 0.6881 |
| | | Xaraés | 0.6659 |
| | | BRS Piatã | 0.5089 |
| | | Mombaça | 0.4427 |
| | | Comum | 0.3386 |
| | | Aruana | 0.2941 |
| | | Tanzânia | 0.0000 |
| 0.7 | 0.3 | ADR 300 | 1.0000 |
| | | Marandu | 0.6734 |
| | | Pojuca | 0.6366 |
| | | Mombaça | 0.5728 |
| | | Xaraés | 0.5183 |
| | | BRS Piatã | 0.3735 |
| | | Comum | 0.1795 |
| | | Aruana | 0.1708 |
| | | Tanzânia | 0.0000 |
| 0.9 | 0.1 | ADR 300 | 1.0000 |
| | | Marandu | 0.6684 |
| | | Mombaça | 0.6214 |
| | | Pojuca | 0.5619 |
| | | Xaraés | 0.4663 |
| | | BRS Piatã | 0.3203 |
| | | Aruana | 0.1044 |
| | | Comum | 0.0839 |
| | | Tanzânia | 0.0000 |

For this reason, it was decided to use an adequate method for multivariate selection (multi-trait). We used the method proposed by De Oliveira et al. [16], which consists of adding the calculation of Manhattan distances in TOPSIS [24]. The latter is a simple but robust method and was designed precisely to solve the type of problem investigated. However, this method cannot be applied directly to the variables measured in the experiments because for many of them, it is not possible to determine whether there is a relationship of the criterion type of maximization (benefit) or minimization (cost). That is, even if the cultivar has suffered the effects of water stress, some variables may increase or decrease their values in relation to those measured in the control environment (see Figs 2 and 3). Therefore, it is not possible to establish that the increase or decrease would characterize a cultivar as more or less susceptible to the aforementioned stress.

The Manhattan distance measurement metrics reflect the similarity between the samples [23]. Thus, if the distance is small, it means that the sample was less influenced by the drought stress environment. Therefore, using the distance measures, we can state that the greater the distance from the control environment, the worse the cultivar [16]. In this way, we can now characterize the criterion for the TOPSIS method as one of minimization.

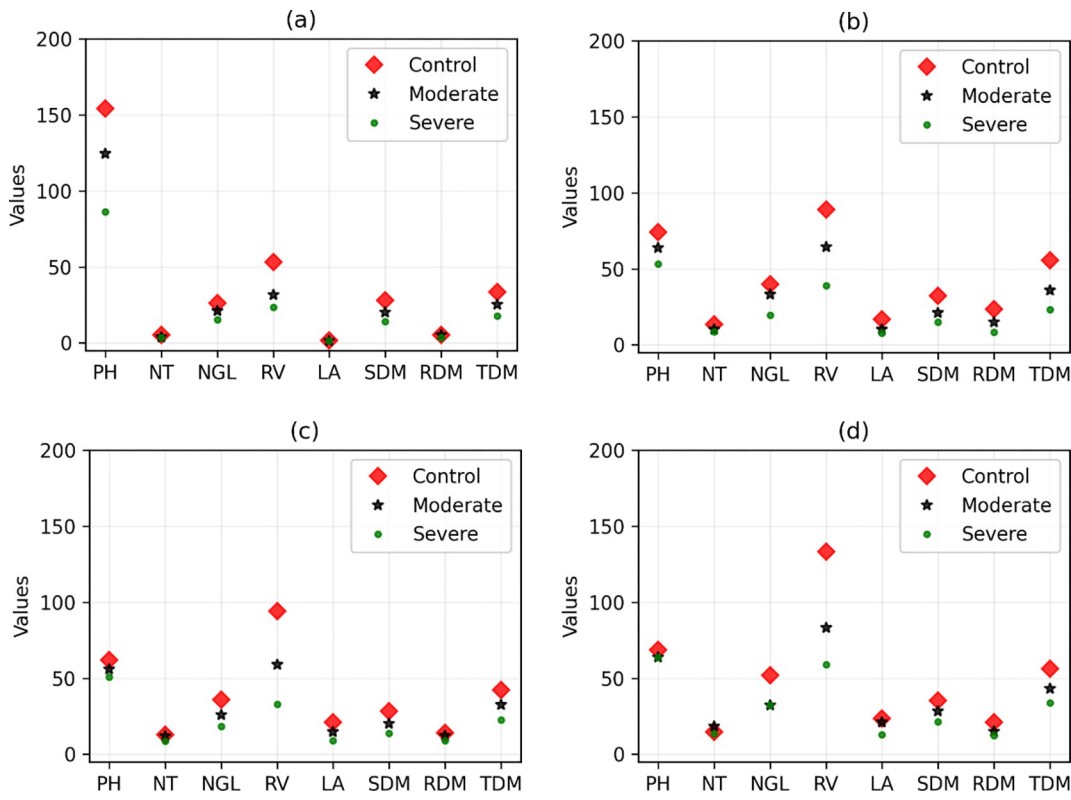

**Fig 2. Values of the variables (without normalization) in the control and drought stress conditions for the four best forage grass varieties.** (a), (b), (c), and (d) correspond to varieties 'ADR 300', 'Pojuca', 'Marandu', and 'Xaraés', respectively. PH: plant height; NT: number of tillers; NGL: number of green leaves; LA: leaf area; SDM: shoot dry matter; RDM: root dry matter; TDM: total dry matter; and RV: root volume.

When evaluating the distance values and the TOPSIS score for each cultivar (Fig 1), it was evident that the variation was significant for all tested cultivars. The *Pennisetum glaucum* cv. 'ADR 300' had the highest score among the TOPSIS scores, and the distances in the stress environments (moderate and severe) were also the lowest. This means that this cultivar is absolutely better than the others because their morphological characteristics suffered much fewer changes in relation to the control environment. In. [7], when employing another technique, stress tolerance indices were used to assess tolerance to water stress, and the authors reported that the cultivar *Panicum maximum* cv. 'Mombaça' has greater adaptability and stability in the production of shoot biomass when cultivated under conditions of water stress. In this study, 12 water stress tolerance indices proposed by several researchers were used to evaluate the forage production response (i.e., shoot biomass production). In all drought stress tolerance indices, shoot biomass production was used for calculation, and *Panicum maximum* cv. 'Mombaça' was the cover crop with the highest values for this variable, both in the control and in the drought environment stress (moderate and severe). The *Pennisetum glaucum* cv. 'ADR 300' had shoot biomass of 28 g, 20.1 g and 14.2 g for the control, moderate and severe stress, respectively. In turn, *Panicum maximum* cv. 'Mombaça' obtained shoot biomass of 41 g, 29 g and 21.6 g for the control, moderate and severe stress, respectively. Thus, when using the stress tolerance indices, species that produce higher production of shoot biomass in both water conditions present better performance when using these 12 indices. On the other hand, the presented method (using the Manhattan distance and TOPSIS) considers only the modifications

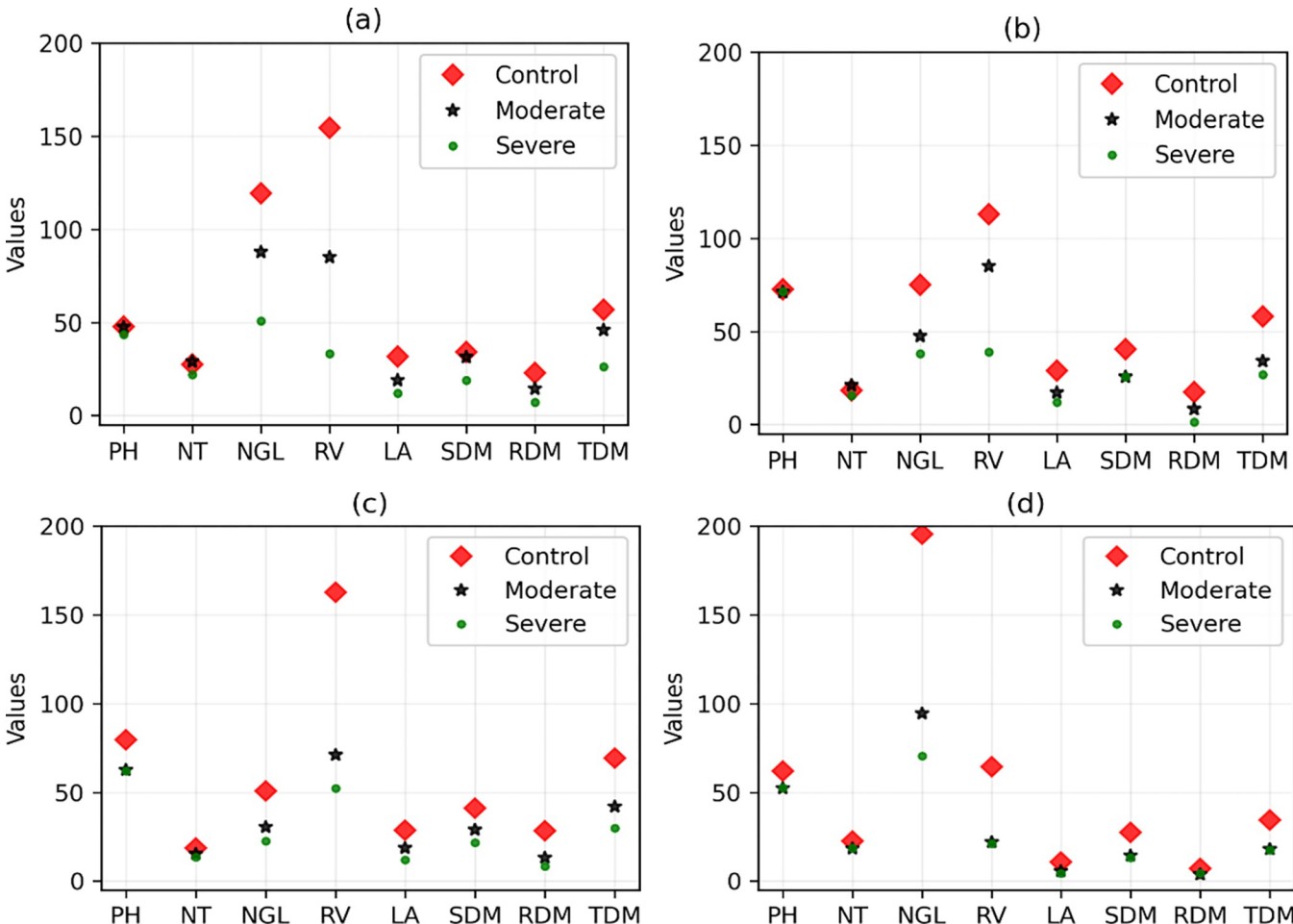

**Fig 3. Values of the variables (without normalization) in the control and drought stress conditions for the four worst forage grass varieties.** (a), (b), (c), and (d) correspond to varieties 'Tanzânia', 'Aruana', 'Comum', and 'BRS Piatã', respectively. PH: plant height; NT: number of tillers; NGL: number of green leaves; LA: leaf area; SDM: shoot dry matter; RDM: root dry matter; TDM: total dry matter; and RV: root volume.

of the variables measured in the stress environments in comparison with those measured in the control environment. That is, this method does not privilege any of the variables, contrary to the method that uses the mentioned indices. Therefore, the presented methods selected different cultivars.

On the other hand, *Panicum maximum* cv. 'Tanzânia' obtained the worst score and the highest distance values. Thus, it is evident that the TOPSIS method makes the selection proper. This can also be confirmed by analyzing the individual values in Table 2, where it is easily seen that the values in the drought stress environments for the 'Tanzânia' cultivar changed more, taking due proportions.

In the previous sections, it was emphasized that the weights of the attributes can change the selection by TOPSIS. Therefore, TOPSIS scores as the weights vary, as shown in Table 4. We note that the 'ADR 300' cultivar was in first place regardless of the weight values. On the other hand, for the other positions, there was a change depending on the weight chosen for the criterion. However, the best and worst cultivars did not change their positions. This could be observed because for these varieties, the differences in distances are more accentuated than for the other varieties.

**Table 5. Percentage change of variable values over the control environment to water stress environments.**

| Cultivar | Comparison | Percentage increase/decrease in variables (%) | | | | | | | |
|---|---|---|---|---|---|---|---|---|---|
| | | PH (cm) | NT (unit) | NGL (unit) | RV (mm³) | LA (cm²) | SDM (g) | RDM (g) | TDM (g) |
| ADR 300 | Control/Moderate | 19.15 | 23.91 | 19.49 | 40.62 | 15.22 | 27.97 | 3.42 | 24.01 |
| | Control/Severe | 44.02 | 30.43 | 41.52 | 56.25 | 18.96 | 49.39 | 33.88 | 46.89 |
| Pojuca | Control/Moderate | 15.59 | 19.22 | 51.64 | 65.71 | 49.20 | 47.65 | 48.27 | 47.78 |
| | Control/Severe | 15.05 | 17.73 | 64.01 | 67.24 | 59.09 | 52.39 | 39.77 | 49.80 |
| Marandu | Control/Moderate | 18.08 | 11.36 | 23.65 | 42.08 | 15.85 | 27.06 | 35.70 | 30.38 |
| | Control/Severe | 20.21 | 36.36 | 41.12 | 64.58 | 52.66 | 50.01 | 61.27 | 54.33 |
| Xaraés | Control/Moderate | 9.67 | 6.95 | 28.39 | 37.35 | 29.50 | 28.74 | 12.85 | 23.49 |
| | Control/Severe | 18.27 | 32.17 | 49.07 | 65.01 | 57.88 | 51.31 | 37.32 | 46.69 |
| Tanzânia | Control/Moderate | 7.28 | -24.81 | 38.11 | 37.50 | 9.41 | 20.21 | 28.36 | 23.24 |
| | Control/Severe | 7.28 | 9.02 | 38.11 | 55.83 | 44.76 | 39.34 | 41.45 | 40.13 |
| Aruana | Control/Moderate | 2.29 | -14.84 | 37.10 | 24.77 | 41.16 | 36.62 | 52.70 | 41.48 |
| | Control/Severe | 1.83 | 14.54 | 49.59 | 65.58 | 58.66 | 36.86 | 92.84 | 53.76 |
| Comum | Control/Moderate | 0.69 | -4.63 | 26.55 | 44.96 | 40.25 | 7.42 | 36.88 | 19.26 |
| | Control/Severe | 9.09 | 20.56 | 57.48 | 78.48 | 62.29 | 44.01 | 69.12 | 54.10 |
| BRS Piatã | Control/Moderate | 13.96 | 24.59 | 16.99 | 27.50 | 39.70 | 34.99 | 35.63 | 35.26 |
| | Control/Severe | 27.92 | 36.06 | 50.69 | 56.25 | 55.02 | 54.10 | 64.80 | 58.59 |

PH: Plant height; NT: Number of tillers; NGL: Number of green leaves; LA: Leaf area; SDM: Shoot dry matter; RDM: Root dry matter; TDM: Total dry matter; and RV: Root volume.

It is important to note that this is not a drawback of the method used but just an option for certain results to stand out over others. That is, if the selector wants to give more emphasis to those varieties that do better in a severe drought environment, he can increase the weight of this criterion. In this case, the varieties selected in the first row of Table 4 would be the appropriate choice. On the other hand, if the selector intends to select those varieties with the best result in the moderate stress environment, he would choose the varieties from the last row of Table 4.

Analyzing the results of Fig 2 and the complementary results of Table 5, we note that for most morphological characteristics, the 'ADR 300' cultivar undergoes less modification than other varieties. However, for the PH variable, the 'ADR 300' cultivar had the largest change among the four best varieties. Even so, changes in other variables make up for the greater change, causing their distances to be the shortest calculated among all varieties. We note in general for the four best varieties that none of them are absolutely better than another. As we are in an 8-dimensional space, it is not possible to analyze all variables simultaneously. Therefore, the TOPSIS multicriteria method combined with distance calculation is very important for this selection.

Comparing the four worst varieties, according to the results presented in Fig 3 and Table 4, it is not clear why the 'Tanzânia' cultivar is the worst according to TOPSIS. However, when we analyze the percentage variation signs, this selection remains evident. Although in absolute values, the 'Tanzânia' cultivar changed less than the 'BRS Piatã' cultivar, the former had a negative variation. This makes the computed distance much larger. It is impossible to visualize this in a space of size greater than 3. However, this is the expected result when you have a vector whose modulus is greater than the others. That is, in this case, the value of the NT variable in the moderate water stress environment was greater than that in the control environment. On the other hand, the other variables have smaller measures. This results in a vector with a much greater length. The same can be observed for the other varieties.

## Conclusions

The findings of this study suggest that the genotypic stability of nine varieties of tropical forage grasses can be assessed by calculating Manhattan distances and using the TOPSIS method. By comparing plants exposed to moderate and severe drought conditions with those grown in a nonstressful environment (control), we were able to determine the most suitable forage varieties for production environments that experience moderate and severe drought stress. Therefore, the TOPSIS method can be utilized to effectively select the best forage varieties for such conditions.

Our findings indicate that ADR 300 (*Pennisetum glaucum*), Pojuca (*Paspalum atratum*), Marandu (*Urochloa brizantha*), and Xaraés (*Urochloa brizantha*) are the most promising varieties for forage production in tropical regions with drought stress. However, further testing under field conditions is necessary to confirm their potential. In addition, our results suggest, considering the inherent aspects of plant breeding, that these four varieties can be used as parents to obtain drought-resistant genotypes.

## Supporting information

**S1 Data.**
(ZIP)

## Author Contributions

**Conceptualization:** Bruno Rodrigues de Oliveira, Alan Mario Zuffo, Francisco de Alcântara Neto, Marcos Renan Lima Leite.

**Data curation:** Bruno Rodrigues de Oliveira, Marco Aparecido Queiroz Duarte, Alan Mario Zuffo, Fábio Steiner, Alexson Filgueiras Dutra, Nágila Sabrina Guedes da Silva, Eliseo Pumacallahui Salcedo, Luis Morales-Aranibar, Richar Marlon Mollinedo Chura, Roger Ccama Alejo.

**Formal analysis:** Bruno Rodrigues de Oliveira, Marco Aparecido Queiroz Duarte, Alan Mario Zuffo, Fábio Steiner, Jorge González Aguilera, Alexson Filgueiras Dutra, Francisco de Alcântara Neto, Eliseo Pumacallahui Salcedo, Wilberth Caviedes Contreras.

**Funding acquisition:** Bruno Rodrigues de Oliveira, Marco Aparecido Queiroz Duarte, Marcos Renan Lima Leite, Richar Marlon Mollinedo Chura.

**Investigation:** Bruno Rodrigues de Oliveira, Marco Aparecido Queiroz Duarte, Alan Mario Zuffo, Jorge González Aguilera, Francisco de Alcântara Neto, Marcos Renan Lima Leite, Nágila Sabrina Guedes da Silva, Luis Morales-Aranibar, Richar Marlon Mollinedo Chura, Roger Ccama Alejo.

**Methodology:** Bruno Rodrigues de Oliveira, Marco Aparecido Queiroz Duarte, Alan Mario Zuffo, Fábio Steiner, Jorge González Aguilera, Alexson Filgueiras Dutra, Marcos Renan Lima Leite, Nágila Sabrina Guedes da Silva, Luis Morales-Aranibar, Richar Marlon Mollinedo Chura, Roger Ccama Alejo, Wilberth Caviedes Contreras.

**Project administration:** Bruno Rodrigues de Oliveira, Alexson Filgueiras Dutra, Francisco de Alcântara Neto, Marcos Renan Lima Leite, Eliseo Pumacallahui Salcedo.

**Resources:** Bruno Rodrigues de Oliveira, Marco Aparecido Queiroz Duarte, Jorge González Aguilera, Marcos Renan Lima Leite, Eliseo Pumacallahui Salcedo, Luis Morales-Aranibar, Richar Marlon Mollinedo Chura, Roger Ccama Alejo, Wilberth Caviedes Contreras.

**Software:** Bruno Rodrigues de Oliveira, Marco Aparecido Queiroz Duarte, Alan Mario Zuffo, Fábio Steiner, Eliseo Pumacallahui Salcedo, Luis Morales-Aranibar.

**Supervision:** Bruno Rodrigues de Oliveira, Alan Mario Zuffo, Fábio Steiner, Jorge González Aguilera.

**Validation:** Bruno Rodrigues de Oliveira, Marco Aparecido Queiroz Duarte, Alan Mario Zuffo, Fábio Steiner, Jorge González Aguilera, Alexson Filgueiras Dutra, Nágila Sabrina Guedes da Silva, Eliseo Pumacallahui Salcedo, Roger Ccama Alejo, Wilberth Caviedes Contreras.

**Visualization:** Bruno Rodrigues de Oliveira, Marco Aparecido Queiroz Duarte, Alan Mario Zuffo, Fábio Steiner, Jorge González Aguilera, Alexson Filgueiras Dutra, Francisco de Alcântara Neto, Marcos Renan Lima Leite, Nágila Sabrina Guedes da Silva, Eliseo Pumacallahui Salcedo, Luis Morales-Aranibar, Richar Marlon Mollinedo Chura, Wilberth Caviedes Contreras.

**Writing – original draft:** Bruno Rodrigues de Oliveira, Marco Aparecido Queiroz Duarte, Alan Mario Zuffo, Fábio Steiner, Jorge González Aguilera, Alexson Filgueiras Dutra, Francisco de Alcântara Neto, Marcos Renan Lima Leite, Nágila Sabrina Guedes da Silva, Eliseo Pumacallahui Salcedo, Luis Morales-Aranibar, Richar Marlon Mollinedo Chura, Roger Ccama Alejo, Wilberth Caviedes Contreras.

**Writing – review & editing:** Bruno Rodrigues de Oliveira, Marco Aparecido Queiroz Duarte, Alan Mario Zuffo, Fábio Steiner, Jorge González Aguilera, Alexson Filgueiras Dutra, Francisco de Alcântara Neto, Marcos Renan Lima Leite, Nágila Sabrina Guedes da Silva, Eliseo Pumacallahui Salcedo, Luis Morales-Aranibar, Richar Marlon Mollinedo Chura, Roger Ccama Alejo, Wilberth Caviedes Contreras.

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
