## [Decision Letter · Decision Letter 0]

3 Jul 2023

PONE-D-23-17029Selection of forage grasses for cultivation under water limited conditions using Manhattan distance and TOPSISPLOS ONE

Dear Dr. de Oliveira,

Thank you for submitting your manuscript to PLOS ONE. After careful consideration, we feel that it has merit but does not fully meet PLOS ONE’s publication criteria as it currently stands. Therefore, we invite you to submit a revised version of the manuscript that addresses the points raised during the review process.

Dear authors, 

I ask you to heed the suggestions of the reviewers and forward the requested adjustments (all in highlights). 

Yours sincerely,

 Julio

We look forward to receiving your revised manuscript.

Kind regards,

Julio Cesar de Souza, Ph.D.

Academic Editor

PLOS ONE

“The funders had no role in study design, data collection and analysis, decision to publish, or preparation of the manuscript”

Reviewers' comments:

Reviewer's Responses to Questions

**Comments to the Author**

1. Is the manuscript technically sound, and do the data support the conclusions?

Reviewer #1: Partly

Reviewer #2: Partly

Reviewer #3: Yes

2. Has the statistical analysis been performed appropriately and rigorously? 

Reviewer #1: Yes

Reviewer #2: No

Reviewer #3: Yes

3. Have the authors made all data underlying the findings in their manuscript fully available?

Reviewer #1: Yes

Reviewer #2: Yes

Reviewer #3: No

4. Is the manuscript presented in an intelligible fashion and written in standard English?

Reviewer #1: Yes

Reviewer #2: No

Reviewer #3: Yes

5. Review Comments to the Author

Reviewer #1: Dear Authors,

I have carefully read the publication submitted for review. The issues of the reaction of grasses to the conditions of periodic water deficits are very close to my experience. Your approach to this issue is interesting and innovative. The presented method of data analysis and drawing conclusions is worth attention. At the same time, I find numerous deficiencies in substantive issues, the possible correction of which will bring this publication closer to publication.

Here are my detailed comments:

Row # 74 – 77: You gave information concerning drought tolerance only to 2 species used in experiment. But what about drought relations known from literature concerning other grass species used?

Row # 90 – 91: It should be stated more general - not only to tropical forage grasses. The understanding of plant reaction to water stress is crucial to further reading,...

Row # 96 – 97: Multivariate analysis methods are widely used in other disciplines, not only agriculture.

Row # 131: Give detail information concerning: seed sowing amount (g/pot or g/square m), date of sowing, any treatment applied after sowing (i.e. seed covering, watering etc.) before seedling emerged. You should also add information concerning experiment design, i.e. number of pots per cultivar, number of observations per pot etc.

Row # 148 – 149: Where do you know from that this PC values were 'moderate drought' or 'severe drought'?

Row # 149 - 150: Yes, but this is only one example of possible drought - plant development relation. Drought may come also in other plant development stages. Are the effect of drought exactly during 25 days at tillering and stalk elongation significantly critical for further plant development? What was the idea of choosing exactly this time of grass development?

Row # 163: Please, add correct citation to the references of the Kuijper’s leaf numbering system.

Row # 163 – 164: It is necessary to know how many live seeds were sown per pot.

Row # 295 – 298: Figures are very good but it lacks significance of difference between means. Therefore in some case there is no effect of the drought conditions on trait mean values. I do suggest to add some ANOVA tables to express the effect of treatments applied.

Reviewer #2: Comments

Manuscript No. PONE-D-23-17029

Title: Selection of forage grasses for cultivation under water limited conditions using Manhattan distance and TOPSIS

Major comments

1. Overall manuscript needs English improvement.

2. There is no molecular based evidence to highlight the reasons the better variety in water deficit stress. I suggset, molecular based research may highlight more on this way.

3. References should be rechecked and arrange according to the journal instruction

4. No author Contributions mentioned in manuscript.

5. No innovated research found in this research treatment.

Minor Comments:

Comment: Keep in word limits to 300 in abstract section.

Comment:

Line 55: Use appropriate keywords please.

Comments:

Line 59: Please recheck the report year and the figures written? 2023 or 2019?

Comment:

Line 79: delete off-season.

Comment:

Line 103: Rewrite this line short and more understand able.

Comment:

Line 107: Forage plants

Comment:

Introduction portion is very lengthy. Please make it short and write essential about problem that can make more easily understandable to the irrelevant reader of your major subject.

Comment:

Line 202: Make it short please.

Comment:

No statistical analysis of the values or parameters that taken from the plants.

Comment:

Line 341: delete “it”.

Comment:

Line 361: Put space between above ground.

Comment:

Line 363: Put comma next to side.

Comment:

Line 364: Put comma next to sections.

Comment:

Line 369: Shift table 2 at the end of the sentence, as shown in Table 2.

Comment:

Line 406: Put comma next to comparing.

Reviewer #3: The authors of this study described the selection of forage grasses for cultivation under water limited conditions by using Manhattan distance and TOPSIS. Manhattan distance was used as a pre-processing step to normalize the data for each variable which ranges from 0 to 1. Then, TOPSIS scores were calculated and used to select the best performing and worst performing cultivars subjected to drought stress. Based on the TOPSIS scores, ‘ADR300’, ‘Pojuca’, “Marandu’, and ‘Xaraés’ were selected as the most drought tolerant cultivars among all samples. The manuscript has a very good command of English and can be accepted after minor revision of the following comments below:

1. Cite other studies that have used TOPSIS in the evaluation of plant varieties or genotypic selection.

2. Before showing normalized values in control and stressed conditions, please include a table to show

your raw data of the morphological characteristics of 9 grass cultivars subjected to control and

drought stress conditions.

3. As mentioned in the introduction, the data in this paper was obtained in the experiment carried out

by Zuffo et al. (2022) using nine cultivars from five grass species. Zuffo et al. (2022) used canonical

correlation to analyze drought indices as compared to this research which used Manhattan and

TOPSIS to analyze morphological variables. However, using the same data, the study of Zuffo et al.

(2022) found that Panicum maximum ‘Mombaca’ has greater adaptability and stability under water

stress while your study identified Pennisetum glaucum ‘ADR. 300’, Paspalum atratum ‘Pojuca’,

Urochloa brizantha ‘Marandu’ and ‘Xaraés’ as the best varieties with greater adaptability and stability.

As such, there are 2 contrasting results using the same data. Discuss further your varying results in

the discussion (L353–363)?

4. Could you explain what “greater adaptability and stable drought tolerance mean”? Why is ‘Xaraés’

which has a TOPSIS score of 0.6 included among other cultivars with greater adaptability and stable

drought tolerance?

5. The authors alternatively used variety or cultivar in the manuscript. Since these are established

grasses, it is better to use “cultivar” and maintain consistency in the manuscript.

Abstract:

Use single apostrophes, not double, for cultivar names here and elsewhere in the manuscript. (eg.,

‘Pojuca’, instead of “Pojuca”).

“Therefore”, instead of “therefore”, in second to the last sentence.

Other comments:

L63 Replace “(Feltran-Barbieri et al., 2021)” with “(Feltran-Barbieri & Féres, 2021)”.

L73, 85, 90 Replace “Mastalerczuk et al., 2021” with “Mastalerczuk & Borawska-Jarmułowicz, 2021”.

L104 What do you mean with “forage grasses tree genotypes”?

L107–108 Restate the incomplete sentence “forage plants resistant to stress conditions”.

L140–144 Use en dash, not hyphen, to indicate a negative value (eg. Kg–1 for Kg-1)

L164 NL or NGL?

L187 Replace “Aggarwal, Hinneburg, and Keim” with “Aggarwal et al.”

L217 Replace “Tzeng, 2011” with “Tzeng & Huang, 2011”.

L218 Replace “Yadav, 2019” with “Yadav et al., 2019”.

L339 Replace “Tzeng, 2011” with “Tzeng & Huang, 2011”.

L341 Complete the sentence for “The TOPSIS method”.

L348 Replace “(Aggarwal, Hinneburg, and Keim, 2001)” with “(Aggarwal et al., 2002)”.

L354–363 When writing cultivar names, “cv.” is replaced with single apostrophes on cultivar names (eg.

Pennisetum glaucum ‘ADR 300’, instead of Pennisetum glaucum cv. “ADR 300”). Please use

single apostrophes consistently for all cultivar names here and elsewhere in the

manuscript.

L357 Delete the comma (,) after “others”.

L359 Replace “Zufo et al” with “Zuffo et al.”.

L359–363 The phrase “a similar forage species to ADR300” which is referring to Panicum maximum

‘Mombaca’ is obviously redundant and may cause confusion as to what it implies. Please delete

or modify this phrase.

L362 ‘ADR 300’, not ADR30

L389 “choosing analyzing”?

L391–393 Revise the sentence for clarity.

L411–412 Use italics for scientific names and single apostrophe for cultivar names

References:

Follow the journal’s style and format.

Is there a period after doi?

Use an en dash (–), not a hyphen (-), when writing page ranges.

L463–465 The authors were not cited in the manuscript.

6. PLOS authors have the option to publish the peer review history of their article (what does this mean?). If published, this will include your full peer review and any attached files.

Reviewer #1: No

Reviewer #2: No

Reviewer #3: No

---

## [Author Response · Author response to Decision Letter 0]

26 Jul 2023

All suggestions from reviewers and editors were accepted. The changes made are listed in the "Response to Reviewers.docx" file.

---

## [Decision Letter · Decision Letter 1]

12 Sep 2023

Selection of forage grasses for cultivation under water limited conditions using Manhattan distance and TOPSIS

PONE-D-23-17029R1

Dear Dr. de Oliveira,

We’re pleased to inform you that your manuscript has been judged scientifically suitable for publication and will be formally accepted for publication once it meets all outstanding technical requirements.

Kind regards,

Julio Cesar de Souza, Ph.D.

Academic Editor

PLOS ONE

Additional Editor Comments (optional):

That's ok.

Publish!

Reviewers' comments:

Reviewer's Responses to Questions

**Comments to the Author**

1. If the authors have adequately addressed your comments raised in a previous round of review and you feel that this manuscript is now acceptable for publication, you may indicate that here to bypass the “Comments to the Author” section, enter your conflict of interest statement in the “Confidential to Editor” section, and submit your "Accept" recommendation.

Reviewer #1: All comments have been addressed

Reviewer #3: All comments have been addressed

2. Is the manuscript technically sound, and do the data support the conclusions?

Reviewer #1: Yes

Reviewer #3: Yes

3. Has the statistical analysis been performed appropriately and rigorously? 

Reviewer #1: Yes

Reviewer #3: Yes

4. Have the authors made all data underlying the findings in their manuscript fully available?

Reviewer #1: Yes

Reviewer #3: Yes

5. Is the manuscript presented in an intelligible fashion and written in standard English?

Reviewer #1: Yes

Reviewer #3: Yes

6. Review Comments to the Author

Reviewer #1: The study present original research findings. Due to my knowledge results reported in the article has not been previously published elsewhere. All experiments, statistical analyses, and other research procedures met a high technical standard. Additionally, these methods and analyses were described in sufficient detail. Conclusions drawn from the research were presented in a manner that is appropriate and logical. Therefore, findings were well-founded and credible. Concluding, it is ready to be published in PLOS ONE.

Reviewer #3: (No Response)

7. PLOS authors have the option to publish the peer review history of their article (what does this mean?). If published, this will include your full peer review and any attached files.

Reviewer #1: No

Reviewer #3: No

---

## [Editor Report · Acceptance letter]

21 Dec 2023

PONE-D-23-17029R1 

PLOS ONE

Dear Dr. de Oliveira, 

I'm pleased to inform you that your manuscript has been deemed suitable for publication in PLOS ONE. Congratulations! Your manuscript is now being handed over to our production team.

Kind regards, 

on behalf of

Dr. Julio Cesar de Souza 

Academic Editor

PLOS ONE